# Three weeks of rehabilitation improves walking capacity but not daily physical activity in patients with multiple sclerosis with moderate to severe walking disability

**Sandra Kuendig**[1,2], **Jan Kool**[1], **Ashley Polhemus**[3], **Wolfgang Schallert**[1]*, **Jens Bansi**[1], **Roman Rudolf Gonzenbach**[4]

**1** Research Department Physiotherapy, Rehabilitation Centre, Valens, Switzerland, **2** Zurich University of Applied Sciences, School of Health Professions, Institute of Physiotherapy, Winterthur, Switzerland, **3** Epidemiology, Biostatistics, and Prevention Institute, University of Zürich, Zürich, Switzerland, **4** Department of Neurological Rehabilitation, Rehabilitation Centre, Valens, Switzerland

\* wolfgang.schallert@kliniken-valens.ch

## Abstract

### Background

Patients with multiple sclerosis have low levels of physical activity. This is of concern because low activity levels are related to cardiovascular disease, poor walking ability, and reduced quality of life. The aim of this study was to evaluate the impact of rehabilitation on daily physical activity and walking capacity in patients with multiple sclerosis who have moderate to severe walking disability.

### Methods

This exploratory, observational study of 24 patients with multiple sclerosis examined daily physical activity, walking capacity and fatigue before and after 3 weeks of inpatient rehabilitation. Inpatient rehabilitation included physiotherapy (30–60 min, 5 times/week), strength and endurance training (30–45 min, 3–5 times/week), occupational therapy (30 min, 2–3 times/week), and neuropsychological training (30 min, 2 times/week). There were no specific interventions to target daily levels of physical activity.

### Results

Daily physical activity did not change after rehabilitation (physical activity: effect size = –0.23, 95% confidence interval (95% CI) 0.02–0.62). There were significant improvements in walking capacity (Two-Minute Walk Test: effect size = 0.74, 95% CI 0.31–1.16, +17 m, 20.2%) and mobility (Timed Up and Go Test: effect size = 0.65, 95% CI 0.22–1.07, –2.1 s, 14.9%). Motor and cognitive fatigue (Fatigue Scale for Motor: effect size = 0.56, 95% CI 0.14–0.99 and Cognitive Functions: effect size = 0.44, 95% CI 0.01–0.86) improved significantly after rehabilitation.

### Conclusion

Three weeks of rehabilitation improved walking capacity, but not daily physical activity, in patients with multiple sclerosis with moderate to severe walking disability. To increase

**Data Availability Statement:** All relevant data are within the manuscript and its Supporting Information files.

**Funding:** The authors received no specific funding for this work.

**Competing interests:** The authors have declared that no competing interests exists.

physical activity, it may be necessary to add specific behavioural interventions to the rehabilitation programme. The intervention plan should include strategies to overcome personal and environmental barriers.

## Introduction

Multiple sclerosis (MS) is a disease of the central nervous system that results in heterogeneous symptoms and progressive functional deficits [1]. These symptoms and deficits lead to low levels of physical activity (PA) in most patients with MS (PwMS). A secondary analysis of pooled data from 13 studies indicated that PA, quantified using accelerometry, is lower in PwMS compared with healthy persons, and is below recommended levels [2]. A meta-analysis found a reduction in PA by almost one standard deviation (SD) compared with non-diseased populations [3]. As the disease progresses with worsening of symptoms and functional deficits, PA in daily life declines further [4, 5]. These low and declining levels of PA in PwMS are of concern, because they are associated with cardiovascular disease, poor walking ability, fatigue, depression and low quality of life [6].

PwMS are less physically active than the general population. Several barriers to PA have been identified in this population, which can be grouped into personal or environmental barriers [7]. Personal barriers comprise ambulatory disability, fatigue, and depression, which are frequent symptoms and comorbidities in MS. Environmental barriers comprise a lack of accessible facilities, insufficient advice on PA from healthcare professionals, or feelings of social exclusion [7]. There is good evidence that exercise therapy and rehabilitation have positive effects on MS-related symptoms and functional impairments. For example, several meta-analyses have shown that exercise therapy improves walking disability [8–10] and fatigue [11, 12]. It seems reasonable to assume that this would lead to an increase in PA, as personal barriers to PA are reduced. However, in our experience, some patients report being more physically active after rehabilitation, whereas others report that their daily PA remains unchanged, despite improvements in walking capacity and fatigue. We are unaware of any studies that have prospectively assessed the impact of exercise therapy or rehabilitation on PA in these patients. In a prospective observational study, Ehling et al. found that multidisciplinary inpatient rehabilitation in PwMS with moderate to severe walking impairment resulted in improved walking capacity, but not walking performance, measured in steps per day [13]. However, Ehling et al.'s study did not examine PA.

The World Health Organization (WHO) defines capacity as an individual's ability to perform a given task or action in a controlled setting, and performance as the activities performed by an individual on a day to day basis in the context of their own life [14].

The objective of this longitudinal study of PwMS was to evaluate the impact of inpatient rehabilitation on PA. It was hypothesised that rehabilitation would increase PA by improving walking capacity and reducing fatigue. Walking capacity, fatigue and mood were secondary outcome parameters, because they have been implicated as barriers to PA in PwMS.

## Methods

### Design and blinding

An exploratory, observational study was conducted in PwMS referred to the Rehabilitation Centre Valens in Switzerland for inpatient rehabilitation. Sample size calculation found that

22 participants were needed to detect a within-group effect size of 0.6 with a power of 0.8, accepting a type I error probability of 0.05 [15]. Recruitment was between 1 August 2017 and 31 March 2018. Participants were consecutively recruited by telephone 2–4 weeks before rehabilitation. Outcomes were evaluated at five time-points: before rehabilitation (T0), at the beginning (T1) and end (T2) of rehabilitation, one week after rehabilitation (T3), and at follow-up 3 months after inclusion (T4). Participants and clinicians were blinded to the results of accelerometer and clinical measurements throughout the study. The study was approved by the ethics committee (BASEC number 2017–00728), registered at ClinicalTrials.gov (NCT03187847) and conducted in accordance with the principles of the Declaration of Helsinki. All participants were informed about the study procedures and provided informed consent.

## Eligibility and recruitment

Eligible subjects were German-speaking PwMS, aged 18 years or older, with moderate to severe disease severity (Expanded Disability Status Scale (EDSS) 3.0–6.5) [16] and a primary rehabilitation goal of improving mobility, as defined by the International Classification of Functioning, Disability and Health (ICF) [14]. Participants were excluded if they were unable to use the accelerometer, had cognitive deficits interfering with study participation, or had comorbidities, such as musculoskeletal or cardiovascular diseases, that reduced walking ability.

The inclusion procedure consisted of two phases. For provisional inclusion, PwMS who were registered for planned rehabilitation were contacted by telephone by a researcher who checked the inclusion criteria and provided verbal information about the study. After provisional inclusion, a letter was sent to patients with written information about the study, an informed consent form, an accelerometer with instructions, and questionnaires regarding fatigue and mood. Definite inclusion was at the start of rehabilitation, when inclusion criteria and patients' ability to use the sensors were checked.

## Rehabilitation

Inpatient rehabilitation was not affected by participation in the study. Therapy generally included physiotherapy to improve balance and walking ability (30–60 min, 5 times/week), strength training (30–45 min, 3 times/week) and endurance training (30–45 min, 2 times/week). Occupational therapy (30 min, 2–3 times/week) focused on energy management and activities of daily living (ADL). Energy management is a form of cognitive behavioural therapy [17] providing information and focusing on coping strategies to tackle fatigue [18, 19]. Neuropsychological training addressed cognitive deficits (30 min, 2 times/week, including training of impaired functions and learning strategies to compensate for deficits). Therapies were individualised according to rehabilitation goals and available therapy resources. There were no specific interventions to target daily levels of PA.

## Outcomes

**Primary outcome: Physical activity.** PA was evaluated with an accelerometer, the Actigraph GT3X (Actigraph, Pensacola, FL, USA), a lightweight device (27 g, 3.8 × 3.7 × 1.8 cm) reported to have good accuracy compared with other devices [20–22]. Participants wore the accelerometer on an elastic belt around their waist above the hip. Patients were asked to wear the accelerometer after getting dressed in the morning and to take it off before going to bed. Because of considerable between-day variation and differences between weekdays and weekend days, participants were instructed to wear the accelerometer for 7 days during waking

hours [23]. Datasets with a minimum wear time of 10 h per day for at least 4 days were considered valid, consistent with previous studies [24–26]. Step count and time spent in PA [27] were calculated in Actilife (Actigraph, Pensacola, FL, USA).

**Walking capacity and mobility.** Waking capacity and mobility were evaluated at the beginning (T1) and end (T2) of inpatient rehabilitation. Walking capacity was evaluated with the Two-Minute Walk Test (2MWT). Participants were asked to walk as fast and as far as possible back and forth along a 30-m hallway, turning around cones at each end, while using their usual walking aids. Mobility was evaluated with the Timed Up and Go Test (TUG) [28]. Patients were seated in a non-armed chair and were asked to sit-up, walk 3 m, turn around a cone at 3-m distance, walk back and sit back down on the chair. The time needed was recorded.

**Fatigue and mood.** Fatigue and mood were assessed when participants were at home, at the following time-points: T0 (before rehabilitation), T3 (1 week after rehabilitation) and T4 (12 weeks after study inclusion). Fatigue was evaluated with the Fatigue Scale for Motor and Cognitive Functions (FSMC) [29], which comprises 20 items, 10 each for the cognitive and motor subscales. Items are rated on a 5-point Likert scale, ranging from 1 (absolutely disagree) to 5 (absolutely agree). FSMC sub-scores for motor and cognitive fatigue range from 10 to 50, interpreted as "mildly fatigued" if 10–26, "moderately fatigued" if 27–31, and "severely fatigued" if >31. Mood was assessed with the depression scale of the Hospital Anxiety and Depression Scale (HADS). Scores between 0 and 7 on HADS are interpreted as "normal", while scores ≥8 points indicate depression in PwMS [30, 31].

## Data analysis

**Clinical outcome assessment.** SPSS Statistics version 24.0 (SPSS Inc., Chicago, IL, USA) was used for the analyses of patient-reported outcomes of fatigue and mood and measurements of mobility and walking capacity during rehabilitation. Because outcomes did not have a normal distribution, as shown by the Shapiro–Wilk test, Wilcoxon signed-rank tests were used to analyse changes. Effect sizes (ES) $r = Z / sqrtN$ with 95% confidence intervals (95% CIs) $= r \pm 1.96/sqrt(N-3)$ were calculated, where Z is the Z-value from the Wilcoxon signed-rank test. ES were considered small if 0.1–0.3, moderate if 0.3–0.5, and large if >0.5.

**Accelerometer-based measurements.** Accelerometer data were analysed using R (v3.6.1) [32]. Due to technical issues, wear-time validation was not available for most recordings in the current study. During recording, all data were previously aggregated at the hourly level. Thus, a custom cut-point-based wear-time validation method was developed. Receiver operating curve (ROC) analysis, theoretical knowledge, visual inspection, and triangulation were used to propose a suitable cut-point. If less than 5 min of wear-time was recorded in a 60 min period, 1 h of the day was considered as non-wear-time. This is consistent with a common non-wear-time definition of 60 min of continuous zeros [33, 34]. Four wear-time validated datasets were available to determine a suitable cut-point, comprising 28 days (648 h) of data (not all days were full days). These data were randomly divided into testing (20%) and training (80%) sets. Ultimately a cut-point of 200 vector magnitude (VM) counts per h was chosen as a theoretically sound option with high specificity (0.99) and sensitivity (0.92). Non-wear-time was filtered from the dataset. Step count and time spent in PA during waking hours (06.00 h to 24.00 h) was aggregated into daily totals. A cut-point of 100 counts per min was used to delineate sedentary behaviour from PA [35]. Distributions and ranges of PA metrics were inspected visually.

Wilcoxon signed-rank test and ES were used to assess the effects of rehabilitation on daily PA, as described for the clinical outcome measures. Mean daily PA was calculated by dividing

total time spent in PA by the number of valid days in the measurement period. In a previous study, PA in healthy persons was significantly different on weekdays compared with weekend days [36], but not in PwMS. Differences in PA between weekdays and weekend days were assessed through univariate and adjusted linear regression. Linear regression models were used to adjust for the confounding effects of walking disability severity, weekends, baseline PA, age, and sex. Walking disability severity was defined as a categorial variable describing either mild/moderate disability (EDSS < = 5) or severe disability (EDSS >5). In a sensitivity analysis, multi-level models were used to account for patient-level random effects.

## Results

### Participants

Participants were recruited between 1 September and 31 October 2017. After the initial telephone contact, 28 PwMS were preliminarily included, of whom 24 were definitely included at the start of rehabilitation. Fig 1 gives an overview of the patient flow in the study and reasons for exclusion.

Baseline characteristics of the 24 participants are reported in Table 1. Data were complete for baseline and outcome measurements. All participants wore the accelerometer at each of the three time-points. Participants had a broad variety of disease duration and severity, mobility, fatigue and depressive mood. Median cognitive fatigue was moderate, while median motor fatigue was severe. Baseline visits took place in September and October 2017, post-rehabilitation assessments were performed in October and November 2017, and the 3-month follow-up occurred between December 2017 and early February 2018.

### Outcomes

Outcomes are reported in Table 2. There was no difference in daily PA at home after, compared with before, rehabilitation. Three months after rehabilitation, the PA was reduced, compared with before rehabilitation. In contrast, walking capacity and mobility evaluated at the beginning and end of 3-weeks' rehabilitation were significantly improved. Self-reported motor and cognitive fatigue and mood were significantly improved at 1 week and at 3 months follow-up after rehabilitation. Fig 2 gives information about interventions and outcome measurements.

**Physical activity.** All participants met minimum wear-time requirements at all time-points. Participants wore the devices for a mean of 14.9 h (SD 1.5) on 6.5 days (SD 8.2) during recording periods. At baseline, participants spent a median of 291 min (interquartile range (IQR) 183–327 min) in PA daily (Fig 3). Time spent in PA was 30 min less (95% CI 6–53, $p = 0.016$) during weekend days compared with weekdays. All except two recordings captured at least one weekend day. Weekends, walking disability severity, age, sex, and baseline PA were treated as potential confounders and adjusted for in subsequent models.

No changes in PA were observed immediately following rehabilitation. However, adjusted linear regression models revealed that participants spent less time in PA at the 3-month follow-up compared with baseline ($p = 0.0055$), representing a median reduction of 30 min (IQR –41 to +3.8) or 9.2% (IQR –21.1% to +1.1%). In all cases, sensitivity analyses accounting for participant-level random effects yielded similar results. At the individual level, most participants (18 of 24) decreased their PA between baseline and the 3-month follow-up, with 11 exhibiting decreases of 20% or more. These 11 participants had significantly higher EDSS than the rest of the study population (median [Q2–Q3]: 6 [6–6.5] vs 4.75 [4–6], $p = 0.011$).

**Walking capacity and mobility.** Walking capacity, evaluated with the 2MWT, and mobility, evaluated with the TUG, improved significantly from start to end of rehabilitation (Table 2).

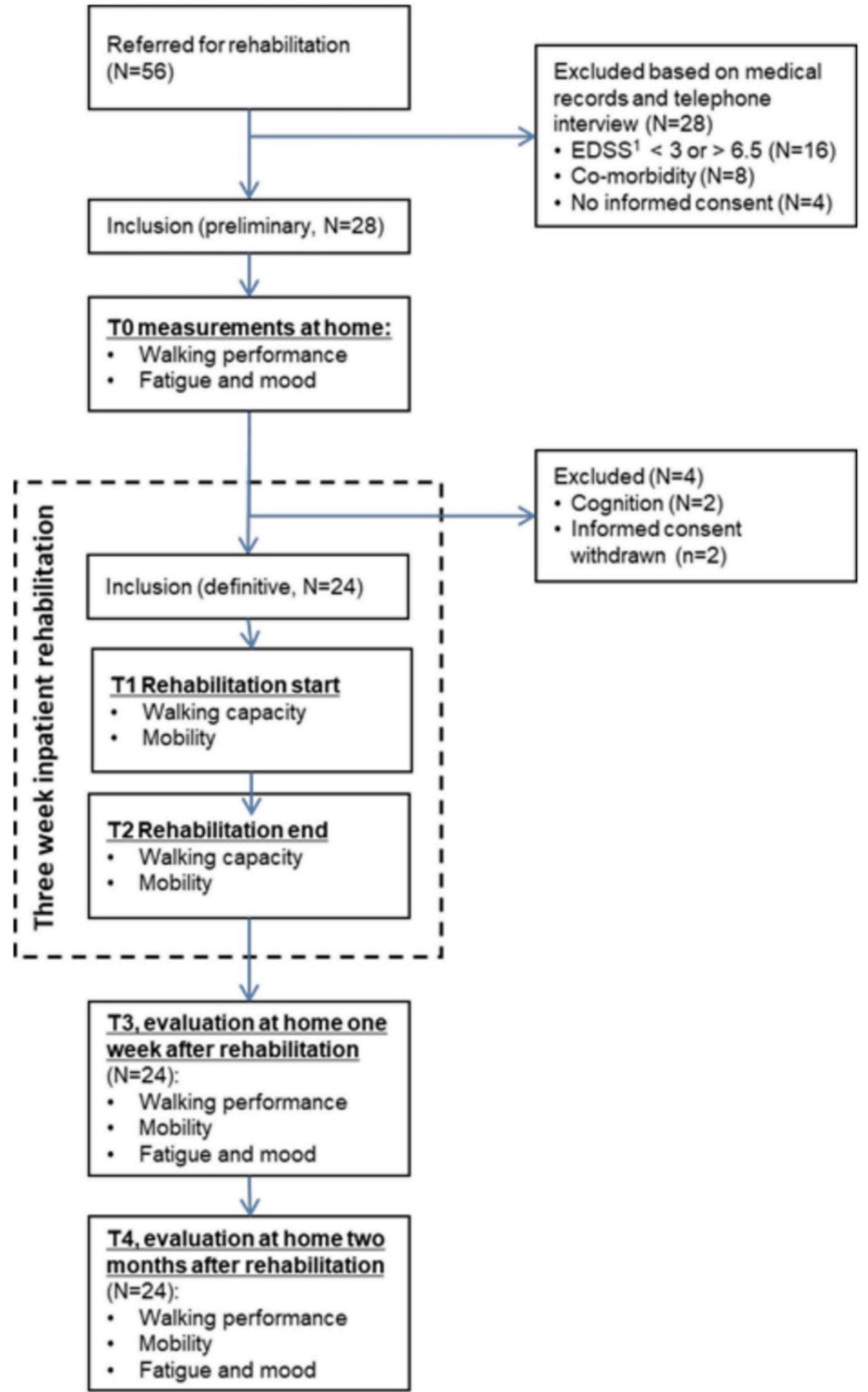

**Fig 1. Study flow chart.** [1]EDSS: Expanded Disability Status Scale.

**Table 1. Baseline characteristics of the 24 participants.**

| Variables | |
|---|---|
| Age, years, mean (SD) | 50.8 (11.1) |
| Sex (female, %) | 12 (50) |
| MS duration, years, median [IQR] | 13.0 [4.8; 17.0] |
| Disease severity, EDSS, median [ÏQR] | 6.0 [4.5; 6.5] |
| Type of MS (*n*) | |
| • primary progressive | 6 |
| • secondary progressive | 8 |
| • relapsing remitting | 10 |
| Mobility TUG, s, median [IQR] | 14.2 [8.9; 20.0] |
| Walking capacity 2MWT, m, median [IQR] | 84.5 [51.5; 125.0] |
| Fatigue FSMC, median [IQR] | |
| • total | 67 [54; 82] |
| • motor | 38 [33; 43] |
| • cognitive | 26.5 [19; 41] |
| Depressive mood, HADS subscale, median [IQR] | 4.0 [3.0; 11] |

MS, multiple sclerosis; EDSS, Expanded Disability Status Scale; TUG, Timed Up and Go test; 2MWT, Two-Minute Walk Test; FSMC, Fatigue Scale for Motor and Cognitive functions; HADS, Hospital Anxiety and Depression Scale.

**Fatigue and mood.** Compared with baseline before rehabilitation (T0), participants reported significantly less fatigue and depressive mood 1 week after rehabilitation (T3) and at 3 months' follow-up (T4), with the exception of cognitive fatigue at T4 (Table 2).

## Discussion

Three weeks of multidisciplinary inpatient rehabilitation in PwMS with moderate to severe walking impairment (EDSS median: 6.0, range: 3–6.5) improved mobility (TUG –2.1 s, –14.9%, $p = 0.002$), walking capacity (2MWT +17 m, +20.2%, $p = 0.002$), fatigue (FSMC –4, –6.3%) and mood (HADS –0.5, –12.5%), but not daily PA, immediately after rehabilitation. At 3 months' follow up, motor fatigue and mood were still improved, but PA had declined, compared with before rehabilitation. The improvements seen in the 2MWT (+17m) and TUG (–2.1 s) exceed the reported minimum for clinically meaningful changes for the 2MWT (+9.1 m) [37] and the TUG (–0.75 s) [38] and are in the range of other rehabilitation studies in MS reporting improvements of 14.8 m (35.1%) [13] for the 2MWT and –1.2 s (–7.9%) for the TUG [39]. The improvement in fatigue (FSMC) by –6 points (–9.4%) at the 3-month follow-up is in line with other studies, which showed a decrease in fatigue levels after rehabilitation [40] of –6.1 points (–12.4%) evaluated with the Modified Fatigue Impact Scale. However, the 6 points (95% CI 2–11) improvement in fatigue on the FSMC 3 months after rehabilitation compared with 1 week before rehabilitation in the current study is below the reported minimum for a clinical meaningful change of 9 points (95% CI –6.8–11.2) reported by Svenningsson et al. [41].

The results of the current study are in line with those of a similar study by Ehling et al. [13], who examined the impact of 28 days of rehabilitation on walking performance, measured in steps per day. The authors reported similar improvements in walking capacity (2MWT +14.8 m) as in the current study, but unchanged steps per day in the subgroup of PwMS with moderate to severe walking impairment, corresponding to an EDSS between 4.0 and 6.5. Although steps per day and PA are different measures of daily life behaviour, the results point in the

**Table 2. Physical activity, walking capacity, mobility, and mood before and after rehabilitation and at 3 months' follow-up.**

| Outcome measure | T0: before rehabilitation | T1: start of rehabilitation | T2: end of rehabilitation | T3: 1 week after rehabilitation | ES vs first measurement [95% CI], p-value[a] | T4: at home at 3 months' follow-up | ES vs first measurement [95% CI], p-value |
|---|---|---|---|---|---|---|---|
| Physical Activity | 291 | | | 267 | -0.23 | 262 | -0.44 |
| Min per day, median [IQR] | [183–327] | | | [202–328] | [0.02; 0.62] | [169–325] | [0.10; 0.81] |
| | | | | | p = 0.23 | | p = 0.029 |
| Walking capacity | | 84 | 101 | | 0.74 | | |
| 2MWT, m, median [IQR] | | [51; 125] | [66; 170] | | [0.31; 1.16] | | |
| | | | | | p = 0.002 | | |
| Mobility | | 14.1 | 12.0 | | 0.65 | | |
| TUG, s, median [IQR] | | [8.9; 20.0] | [5.8; 15.3] | | [0.22; 1.07] | | |
| | | | | | p = 0.002 | | |
| Motor fatigue | 38 | | | 36 | 0.56 | 34 | 0.49 |
| FSMC (10–50 max.), median [IQR] | [33; 43] | | | [30; 38] | [0.14; 0.99] | [27; 40] | [0.07; 0.92] |
| | | | | | p = 0.004 | | p = 0.012 |
| Cognitive fatigue | 26 | | | 24 | 0.44 | 24 | 0.38 |
| FSMC, (10–50 max.), median [IQR] | [19; 41] | | | [17; 36] | [0.01; 0.86] | [18; 35] | [−0.05; 0.80] |
| | | | | | p = 0.033 | | p = 0.065 |
| Depressive mood | 4.0 | | | 3.5 | 0.61 | 3.5 | 0.50 |
| HADS[b], median [IQR⁴] | [3; 11] | | | [1; 9] | [0.18; 1.03] | [2; 7] | [0.07–0.93] |
| | | | | | p = 0.003 | | p = 0.014 |

ES: effect size; r, Wilcoxon test statistic Z/sqrt(n), positive values indicate improvement; 95% CI, 95% confidence interval; IQR, interquartile range; 2MWT, Two-Minute Walk Test; TUG, Timed Up and Go test; FSMC, Fatigue Scale for Motor and Cognitive functions; HADS, Hospital Anxiety and Depression Scale.

[a]p-value of the non-parametric Wilcoxon signed-rank test.

[b]Chi-square test.

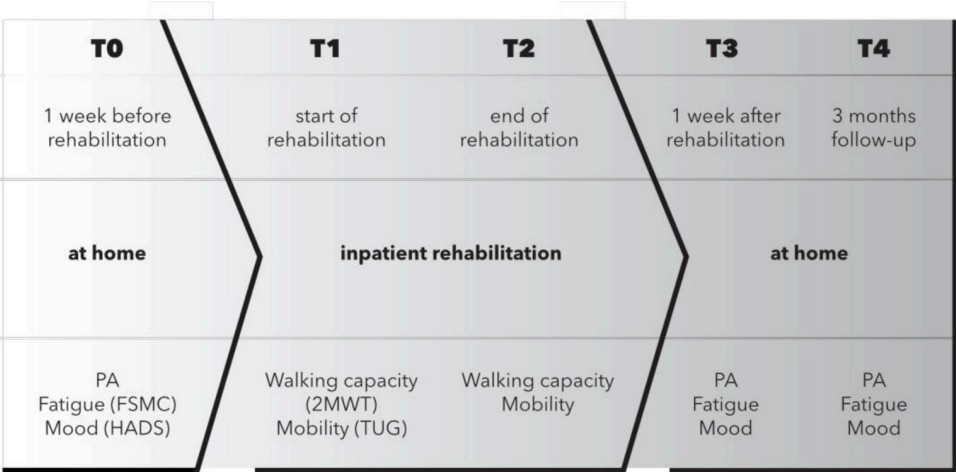

**Fig 2. Time schedule of interventions and outcome measurements.** Abbreviations: PA, physical activity; FSMC, Fatigue Scale for Motor and Cognitive functions; HADS, Hospital Anxiety and Depression; 2MWT, Two-Minute Walk Test; TUG, Timed Up and Go test; T0–4, time point for outcome measurements.

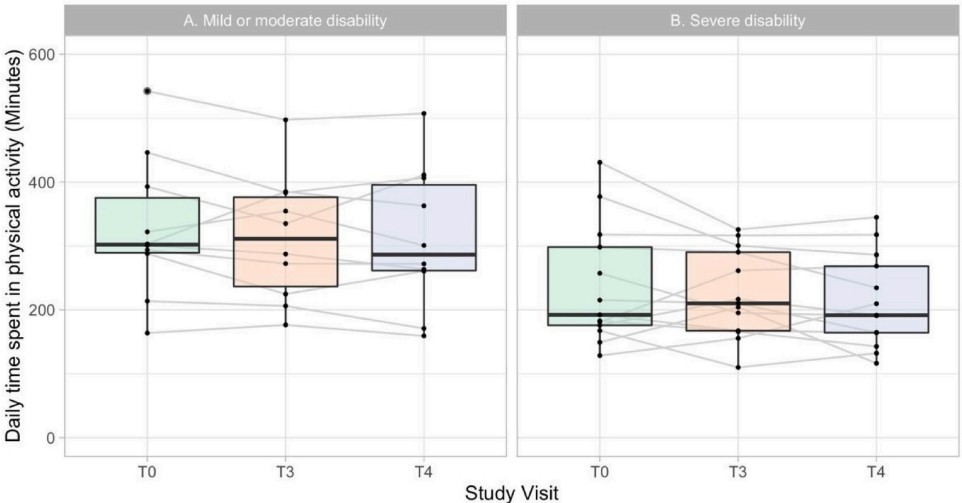

**Fig 3. Median daily physical activity before and after rehabilitation.** (A) Participants with mild to moderate walking disability, defined as Expanded Disability Status Scale (EDSS) less than or equal to 5. (B) Participants with severe walking disability, defined as EDSS greater than 5. T0: the week prior to rehabilitation; T3: the week following rehabilitation, T4: 3-month follow-up.

same direction: rehabilitation resulted in improved walking capacity without altered daily life behaviours, such as steps per day or PA.

This study demonstrates that improvements in mobility and walking capacity do not necessarily translate into increased physical activity behaviour after rehabilitation. One possible explanation is that walking impairment is not the only factor that affects PA in PwMS. Fatigue and depression, which are both common in MS, are further personal barriers to PA [7]. In the current study, the patients reported relatively high levels of fatigue (FSMC 64) and reduced mood (HADS 4.0), and both factors may have prevented higher levels of PA. Both fatigue and mood improved after rehabilitation, but only to a small to moderate degree, which was below clinical relevance [41]. Another explanation might be that environmental factors, such as stairs, uneven surfaces, or ascents/descents, etc., remained unsurmountable barriers for patients with moderate to severe walking deficits, even after improving walking capacity. Such environmental barriers are likely to be more relevant in moderate and severe walking deficits, as in the current study, and less relevant for those with only minor walking deficits. Other environmental barriers, such as inappropriate access for disabled persons, or lack of disabled facilities, cannot be removed by rehabilitation and may have resulted in limitations in PA [7]. Furthermore, the lack of strategies to change behaviour is a barrier to changing PA in PwMS [7].

Further research is warranted to identify strategies and novel components of rehabilitation programmes that will enable the translation of gains in walking capacity into changes in real-world walking behaviours. Although all patients received instructions for an individual home training programme before discharge, the rehabilitation programme in this study did not include a dedicated intervention aimed at increasing PA. Including a dedicated behavioural treatment with the goal of increasing PA as part of the rehabilitation programme might have an effect on PA levels. Behavioural treatment should include goal-setting, patient education, tailored activity planning, addressing self-efficacy, and problem solving [6, 42–44]. Treatment should also identify and address barriers to PA in individual patients, such as fixed personal routines, fatigue, mood, or lack of motivation [45, 46]. Furthermore, internet-based

interventions supplemented by video coaching have been proposed for behavioural treatment [46]. The reason why PA decreased slightly, by 10% at 3 months' follow-up (T4) compared with baseline, is not clear. Seasonal effects may have contributed to this result, as participants were recruited in summer and autumn, and the 3 months' follow-up occurred during winter [47, 48]. Furthermore, it cannot be excluded that disease progression also played a role, even though the follow-up duration was only 3 months [49].

## Study limitations

Since this is an exploratory single-group non-randomised observational study, effect sizes presented in the manuscript do not ascribe causality. Female PwMS were under-represented in the present study (50%) compared with the Swiss population of PwMS (74%) [50]. The patient population was recruited at a single centre, and is of a relatively small sample size, which limits the generalisability of the results. As most patients had moderate to severe disability, the results cannot be generalised to PwMS with mild disability. For technical reasons a non-standard approach to wear time validation was followed, and this may have affected point estimates of time spent in PA. However, we were able to validate our method in a subset of data against traditional wear time validation methods.

## Conclusion

Multidisciplinary inpatient rehabilitation that includes physical therapy, fitness and education in energy management significantly improved walking capacity, mobility, fatigue and mood, but not PA. Further research should examine whether adding goal-directed behavioural interventions to inpatient rehabilitation increases PA in PwMS.

## Supporting information

**S1 Table. Physical activity.** Data is presented as median (min per day). Abbreviations: T0, one week before rehabilitation; T3, one week after rehabilitation; T4, 3-month follow up.
(XLSX)

**S1 Dataset. SPSS dataset of baseline characteristics and clinical outcomes.**
(SAV)

## Author Contributions

**Conceptualization:** Roman Rudolf Gonzenbach.

**Data curation:** Jan Kool, Ashley Polhemus, Wolfgang Schallert.

**Formal analysis:** Jan Kool, Ashley Polhemus, Wolfgang Schallert.

**Investigation:** Sandra Kuendig.

**Methodology:** Jan Kool.

**Project administration:** Jan Kool.

**Resources:** Jens Bansi.

**Supervision:** Roman Rudolf Gonzenbach.

**Writing – original draft:** Sandra Kuendig.

**Writing – review & editing:** Wolfgang Schallert, Roman Rudolf Gonzenbach.

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
