## [Decision Letter · Decision Letter 0]

11 Jul 2022

PONE-D-22-10453Three weeks of rehabilitation improves walking capacity without altering daily physical activity in patients with multiple sclerosis with moderate to severe walking disabilityPLOS ONE

Dear Dr. Schallert,

Thank you for submitting your manuscript to PLOS ONE. After careful consideration, we feel that it has merit but does not fully meet PLOS ONE’s publication criteria as it currently stands. Therefore, we invite you to submit a revised version of the manuscript that addresses the points raised during the review process.

Additional Editor Comments (if provided):

The reviewers have commented some revisions on your manuscript. King Regards

We look forward to receiving your revised manuscript.

Kind regards,

Fatih Özden, PhD

Academic Editor

PLOS ONE

Journal Requirements:

a) Did participants provide their written or verbal informed consent to participate in this study?

b) If consent was verbal, please explain i) why written consent was not obtained, ii) how you documented participant consent, and iii) whether the ethics committees/IRB approved this consent procedur

3. We note that you have stated that you will provide repository information for your data at acceptance. Should your manuscript be accepted for publication, we will hold it until you provide the relevant accession numbers or DOIs necessary to access your data. If you wish to make changes to your Data Availability statement, please describe these changes in your cover letter and we will update your Data Availability statement to reflect the information you provide

Reviewers' comments:

Reviewer's Responses to Questions

**Comments to the Author**

1. Is the manuscript technically sound, and do the data support the conclusions?

Reviewer #1: Yes

Reviewer #2: Yes

Reviewer #3: Partly

2. Has the statistical analysis been performed appropriately and rigorously? 

Reviewer #1: Yes

Reviewer #2: Yes

Reviewer #3: No

3. Have the authors made all data underlying the findings in their manuscript fully available?

Reviewer #1: No

Reviewer #2: Yes

Reviewer #3: Yes

4. Is the manuscript presented in an intelligible fashion and written in standard English?

Reviewer #1: Yes

Reviewer #2: Yes

Reviewer #3: Yes

5. Review Comments to the Author

Reviewer #1: MS number: PONE-D-22-10453

Review Comments to the Author:

The authors have conducted an exploratory, observational study that measured the impact of a 3-week inpatient rehabilitation that included physiotherapy, strength and endurance training, occupational therapy, and neuropsychological training, on daily physical activity levels, walking capacity, and fatigue on 24 patients with moderate to severe Multiple Sclerosis (MS).

It is interesting to note that the standard components of MS rehabilitation when conducted inpatient for 3 weeks improved walking capacity, mobility, and motor and cognitive fatigue. As expected, the inpatient rehabilitation for 3 weeks did not have an effect on the physical activity levels after discharge, as there were no specific interventions to improve daily physical activity levels post-discharge.

I notice that you have adjusted your regression models for disease severity (line 190) (or walking disability severity – line 232), and other potential confounders. Can you pls clarify whether you have used total EDSS scores or any other specific walking disability measure for these analyses?

Some researchers debate that the difference in disability status between EDSS scores are not equal (for e.g., EDSS 3.0 to 3.5 vs 6.0 to 6.5 represent no vs large change in walking disability). I see that the inter-quartile range for EDSS scores at baseline were 4.5 to 6.5. Does this mean that there were participants with no walking impairment (EDSS 3.0 to 3.5) in your study? If so, I am thinking whether such a limitation of EDSS scale could have had a confounding effect on your analyses. Pls acknowledge this as a limitation, and/or if necessary, pls revise your analyses.

It is nice to read that some of your results are in line with the literature, as per your citations in the discussion. Are you able to report a comparison with adequate data/explanation, instead of textual statements (as in lines 256-7)?

Since this is an exploratory single group non-randomized observational study, pls acknowledge that the effect sizes presented in the manuscript do not ascribe causality, in your limitation section.

I see that there were 50% females in your sample (table 1). Pls acknowledge this as a limitation as it does not represent male to female ratio of MS prevalence.

In the introduction, you have mentioned that, in your experience, patients with MS report having either more or no change in physical activity levels after rehabilitation. I notice that the lower end of inter-quartile range for physical activity level has decreased from 202 to 169 min/day between T3 and T4 time points (table 2). Have you considered describing the characteristics of those who had a decrease in their physical activity levels after rehabilitation through a sub-group analysis? Or, you may consider presenting the spread of your individual data points (connected/paired pre-post) using figures with/without error bars.

I acknowledge that relevant statistical analyses were completed to support the conclusions. However, there is no mention of how missing data were handled (other than accelerometer data), if any.

Reviewer #2: Thank you for the opportunity to review this manuscript. I have read it with great interest.

The authors performed a study on the effects of inpatient rehabilitation on physical activity, walking capacity and fatigue in patients with MS.

I have reviewed the well-conducted and well-reported manuscript and would like to make some minor suggestions on how to improve the quality of reporting.

Title & Abstract:

1. Please avoid abbreviations. Especially FSMCc and FSMCm are not explained.

2. Please avoid p values according to recent recommendations. Rather report 95% CIs. Please refer to doi: 10.1016/j.physio.2021.12.003

Introduction/Background:

3. A “question” such as “Why are PwMS less physically active than the general population?” seems uncommon for a scientific report and may be revised. Eg, “PwMS are less physically active than the general population because of xyz”.

Methods:

4. Please give examples for the “comorbidities that reduced walking ability” (exclusion criterium).

5. I suggest to reporting the recruitment and data collection periods in the methods section (if defined a priori before the data collection).

6. Please provide a sample size calculation or justification/rationale.

Results:

7. Well reported.

Discussion:

8. Please state the exact values of the “reported minimum for clinically meaningful changes” used in this study ands exceeded by the participants.

9. Line 264: What is the meaning of the EDSS values (EDSS 4.0–6.5)? Range, IQR?

10. Please avoid questions such as in line 268 and 289.

11. Please explain why the heterogeneous population of PwMS is a limitation.

12. I suggest to discuss the issues with data collection (“Due to technical issues, wear time validation was not available for most recordings in the current study.”) in the limitations section.

13. Since this is an explorative study, generalisability is not the focus and thus, this is not a weakness of the study. The authors may clearly state this and rather discuss future directions for further research and how this study might inform further studies with a bit more passion.

I hope that these comments will help the authors to further improve the manuscript. I would be happy to read a revision of the manuscript.

Sincerely, Tobias Braun

Reviewer #3: The purpose of this study is to assess the effect of rehabilitation on daily physical activity and walking ability in multiple sclerosis patients with moderate to severe walking disability. This study, I believe, is an insightful examination that offers light on an essential problem. However, the author must address a number of critical issues before publication.

I have compiled the following list of criticisms that the authors need to address:

1. Why some exercises were given 5 times per week while others 3-5 or 2-3 times/week?

2. How physiotherapy and strength/endurance training are different? What type of interventions were given in physiotherapy? please clarify

3. What are the activities included under Neuropsychological training?

4. Did you assess the reliability of each outcome measure used in the current study?

5. did you find any variations in test-retest reliability of any outcome measure?

6. Please provide details of sample size estimation.

7. Follow-up periods were inconsistent. For example, in the beginning it was mentioned 3 months, however, in page 11 it was 2 months? please clarify

8. Please include some figures showing interventions and findings for better understanding and interpretation.

Thank You

6. PLOS authors have the option to publish the peer review history of their article (what does this mean?). If published, this will include your full peer review and any attached files.

Reviewer #1: No

Reviewer #2: No

Reviewer #3: No

---

## [Author Response · Author response to Decision Letter 0]

2 Aug 2022

Thank you for your positive feedback on our manuscript, and for the extensive suggestions for improvements, which we have copied in this letter. We have noted our changes and comments in italics below, based on your suggestions. We hope that we have addressed all of the comments to your satisfaction.

The authors.

---

## [Decision Letter · Decision Letter 1]

26 Aug 2022

Three weeks of rehabilitation improves walking capacity but not daily physical activity in patients with multiple sclerosis with moderate to severe walking disability

PONE-D-22-10453R1

Dear Dr. Schallert,

We’re pleased to inform you that your manuscript has been judged scientifically suitable for publication and will be formally accepted for publication once it meets all outstanding technical requirements.

Kind regards,

Fatih Özden, PhD

Academic Editor

PLOS ONE

Additional Editor Comments (optional):

Reviewers' comments:

Reviewer's Responses to Questions

**Comments to the Author**

1. If the authors have adequately addressed your comments raised in a previous round of review and you feel that this manuscript is now acceptable for publication, you may indicate that here to bypass the “Comments to the Author” section, enter your conflict of interest statement in the “Confidential to Editor” section, and submit your "Accept" recommendation.

Reviewer #1: All comments have been addressed

Reviewer #2: All comments have been addressed

Reviewer #3: All comments have been addressed

2. Is the manuscript technically sound, and do the data support the conclusions?

Reviewer #1: Yes

Reviewer #2: Yes

Reviewer #3: Yes

3. Has the statistical analysis been performed appropriately and rigorously? 

Reviewer #1: Yes

Reviewer #2: Yes

Reviewer #3: Yes

4. Have the authors made all data underlying the findings in their manuscript fully available?

Reviewer #1: No

Reviewer #2: Yes

Reviewer #3: Yes

5. Is the manuscript presented in an intelligible fashion and written in standard English?

Reviewer #1: Yes

Reviewer #2: Yes

Reviewer #3: Yes

6. Review Comments to the Author

Reviewer #1: Thank you for addressing all of my concerns and suggestions.

The PA_Data.xlsx file is password protected and the supporting information renamed_f3a4d.sav file is not readable.

Reviewer #2: Thanks for the invitation to review this revises manuscript. All my comments have been satisfactorily addressed by the authors and I have no further comments.

Reviewer #3: Authors have revised the manuscript based on the reviewer comments. Manuscript is much improved now. No further comments.

7. PLOS authors have the option to publish the peer review history of their article (what does this mean?). If published, this will include your full peer review and any attached files.

Reviewer #1: No

Reviewer #2: No

Reviewer #3: No

---

## [Editor Report · Acceptance letter]

9 Sep 2022

PONE-D-22-10453R1 

Three weeks of rehabilitation improves walking capacity but not daily physical activity in patients with multiple sclerosis with moderate to severe walking disability 

Dear Dr. Schallert:

I'm pleased to inform you that your manuscript has been deemed suitable for publication in PLOS ONE. Congratulations! Your manuscript is now with our production department. 

Kind regards, 

on behalf of

Dr. Fatih Özden 

Academic Editor

PLOS ONE